# The Role of Melatonin in Pregnancy and the Health Benefits for the Newborn

**DOI:** 10.3390/biomedicines10123252

**Published:** 2022-12-14

**Authors:** Rosita Verteramo, Matteo Pierdomenico, Pantaleo Greco, Carmelia Milano

**Affiliations:** Department of Obstetrics and Gynecology, Azienda Ospedaliero-Universitaria S. Anna, University of Ferrara, Cona, 44122 Ferrara, Italy

**Keywords:** melatonin gestational period, melatonin in pregnancy, melatonin induction of labor, melatonin fetal health, melatonin and circadian rhythm, melatonin during parturition, melatonin labor, melatonin in breastfeeding, melatonin neuroprotection, melatonin in preeclampsia, melatonin in gestational diabetes, melatonin in fetal growth restriction

## Abstract

In the last few years, there have been significant evolutions in the understanding of the hormone melatonin in terms of its physiology, regulatory role, and potential utility in various domains of clinical medicine. Melatonin’s properties include, among others, the regulation of mitochondrial function, anti-inflammatory, anti-oxidative and neuro-protective effects, sleep promotion and immune enhancement. As it is also bioavailable and has little or no toxicity, it has been proposed as safe and effective for the treatment of numerous diseases and to preserve human health. In this manuscript, we tried to evaluate the role of melatonin at the beginning of human life, in pregnancy, in the fetus and in newborns through newly published literature studies.

## 1. Introduction

Melatonin is a lipophilic hormone synthesized and secreted mainly in the pineal gland, acting as a neuroendocrine transducer of photoperiodic information during the night. In addition to this activity, melatonin has shown an antioxidant function and a key role as a regulator of physiological processes related to human reproduction [1]. Several studies have evaluated its role in ovarian dysfunction [2], ovarian ageing, embryo maturation [3] and gynecological cancer [4]. However, there are little data on the effects of melatonin supplementation during pregnancy and what the potential outcomes are [1]. Some recent research has shed light on the mechanisms of action and functions in the gestational period. Among the physiological activities of this molecule, one of the most important is the modulation of the maternal reproductive circadian rhythm, which favors uterine receptivity, and establishes the day/night cycle of the fetus. Melatonin is also synthesized in other tissues, such as placental tissue and, being able to rapidly pass through the placenta, sends photoperiodic information to the fetus, supporting tissue differentiation and hormonal metabolism [5]. Placental tissue is characterized by a very strong relationship between its two components, cytotrophoblast and syncytiotrophoblast, and plays a key role during implantation and pregnancy. It is believed that there are melatonin functions of induction and blocking of apoptosis that are exploited by the placenta to maintain a balance between the villous cytotrophoblasts and the syncytiotrophoblast, therefore serving to prevent placental pathologies [6]. Considering that melatonin receptors are present at the fetal level, there is evidence that melatonin is involved in neurodevelopment; that is, in the development of fetal sleep. In fact, during the last trimester of pregnancy, faster fetal brain development is closely associated with REM (Rapid Eye Movement) sleep [7]. Finally, a recent randomized double-blind study (MILO) has considered the role of melatonin in the early active phase of induced labor, since the amount of melatonin, which increases in full-term pregnancy reaching a peak during labor, amplifies the sensitivity of myometrial receptors to oxytocin [8,9]. The studies cited in this manuscript are of use to review the existing literature submitted within the last 5–10 years regarding the function of melatonin in pregnancy, labor and in the first period after delivery to increase the prospect of using melatonin as a therapeutic treatment (Table 1). We selected the articles based on the question “What is the role of melatonin for pregnancy and the benefit of the newborn?”

In Table 1 we have summarized in chronological order the main studies used in our review, the study designs varied over time in terms of case-control, review, peer-review and randomized controlled trial. Table 1 structured in order to facilitate reading for evaluation the types of studies, purposes of the studies and the main outcomes and conclusion. In recent years, studies on melatonin have increased in quantity as well as in quality of studies but there are still no studies that allow us to give certainties on the clinical use of melatonin, in the main guidelines there are no clear indications of the use of melatonin, in studies it represents a very interesting molecule for its anti-inflammatory efficacy against oxidate stress and the formation of free radicals.

## 2. Metabolism

Melatonin, also known as 5-methoxy-N-acetyltryptamine, is a neuroendocrine hormone that is mostly produced in the pineal gland, regulated by light levels and follows a circadian rhythm [10]. It is an endogenously-formed indoleamine, comprising two functional groups: the 5-methoxy group and the N-acetyl side chain, which determine its specificity and amphiphilicity. The melatonin biosynthesis starts with the precursor to melatonin, tryptophan, and consists of four enzymatic steps, but at least six enzymes are known to be involved. Within melatonin biosynthesis, serotonin N-acetyltransferase is the rate-limiting enzyme and serotonin is an essential intermediate [11]. The half-life of melatonin in the circulation is generally short, varying in the range of 30–60 min [12]. It has amphiphilic characteristic, that allows melatonin to easily diffuse and cross all membranes. Approximately 70% of melatonin is bound to albumin, and the remaining 30% diffuses to the surrounding tissues after release into the blood [13]. Melatonin is largely metabolized in the liver and kidneys by P450 monooxygenases, followed by conjugation of the resulting 6-OHMS to produce the main metabolite, 6-sulfatoxymelatonin (aMT6s), that is excreted in the urine [14,15]. Furthermore, melatonin can be metabolized by non-enzymatic pathways: N1-acetyl-N2-formyl-5-methoxykynuramine (AFMK) and N1- acetyl-5-methoxykynuramine (AMK) are two major melatonin-derived kynuramines of cyclic 3-hydroxymelatonin. These metabolites serve to extend the timing of its action because melatonin is rapidly metabolized in the peripheral tissues [16,17]. Melatonin receptors are formed by seven transmembrane domains coupled to G protein, termed melatonin receptor 1 (MT1) (also called MTNR1A or Mel1a), MT2 (also called MTNR1B or Mel1b), and MT3, an intracellular enzyme (identified as quinone reductase II enzyme); they often act as homo or heterodimers to reduce the levels of cAMP and cGMP, thereby increasing the Phosphorylation of the other enzymes (Figure 1) [15,18]. The sleep-promoting effect of melatonin is attributed to the activation of the MT1 receptor in the SCN (suprachiasmatic nucleus), while the phase-shifting activity of melatonin is linked to the MT2 receptor [15]. Through its multiple actions, melatonin works as a circadian regulator, natural antioxidant, and anti-inflammatory and oncostatic agent, in addition to its many other functions [18]. Regarding the anti-inflammatory and circadian regulator functions, the study by Wai Man G.C. et al., 2017 shows the correlation between the circadian clock and the immune system. It is underlined that melatonin is potent in terms of circadian regulation of lymphocyte proliferation, enhancing phagocytosis and stimulating cytokine production. Therefore, in the setting of obstetrics and fetal growth, melatonin exerts potential beneficial effects in suppressing various diseases with inflammatory origins, including preterm labor, gestational diabetes, and preeclampsia [19].

## 3. Melatonin in Pregnancy

Melatonin has an important role in human reproduction, it is essential in each phase of ovulation, fertilization, embryo implantation and finally as a pregnancy regulator [1]. The role of melatonin in the first time of reproduction has been deeply investigated. The study by Carlomagno G. et al., 2018, reviewed several in vitro and in vivo studies, confirming that the antioxidant action by melatonin to remove free radicals from the oocyte and the embryo, together with the immunomodulator action, is essential to support implantation and proliferation. This is demonstrated by the presence of melatonin in higher concentrations in ovary tissue and by MT1 and MT2 receptors expressed in the ovary and placenta tissue during embryo implantation for improving the success and quality of embryo development. Throughout gestation, serum melatonin concentrations in the mother display fluctuations in both physiological and pathological pregnancies [20]. In adults, melatonin levels remain low throughout the day. In the early evening, the levels begin to increase, peaking between 02:00 and 03:00 and then falling back down to low daylight concentrations again in the morning [21]. This production followed the 24 h rhythm and relies on a series of gene transcription and translation feedback loops by a group of circadian genes known as “clock genes”, which are contained in the placenta and that are modulated especially by melatonin hormone and glucocorticoids ones [22]. In fact, the circadian rhythm is amplified during pregnancy likely due to de novo melatonin placental synthesis. Particularly, a significant increase in serum melatonin occurs after 24 weeks after implantation, increasing again after 32 weeks [23]. The relationship between the circadian clock system and the immune system is important during pregnancy. Wai M.G.C. et al., 2017 explained in an animal and human study that immunity and clock gene control is important during pregnancy, because through the modulation in T-cell responses, melatonin exerts potential beneficial effects in suppressing various diseases with inflammatory origins, including preterm labor, gestational diabetes, and preeclampsia (Figure 2). Furthermore, in the third trimester of pregnancy, melatonin crosses the placenta and blood–brain barrier from the maternal circulation to the fetus and melatonin receptors are widespread in the fetus in both central and peripheral tissue from early fetal development [19]. Lastly, after birth, the values return to physiological levels within two days [23]. Another potential application of melatonin could be its use in the treatment of insomnia during pregnancy, but we did not find trials where the primary outcomes suggested the safety or efficacy of melatonin for insomnia or other sleep disorders during pregnancy.

## 4. Melatonin and Placenta

As mentioned previously, melatonin serum concentration is higher in pregnant women due to de novo placental synthesis [23]. In fact, the placenta represents the source of melatonin and it also contains a melatonin receptor. The surface of the placenta includes two different types of cells: mononuclear villous cytotrophoblast and the multinucleated syncytiotrophoblast. The fusion of the cells villous cytotrophoblasts forms the syncytiotrophoblast by the melatonin regulated process. Specifically, the melatonin preserves the apoptosis of villous cytotrophoblast to promote the development of the syncytiotrophoblast due to paracrine, autocrine and/or intracrine actions of the MT1 and MT2 receptors in the placenta [24]. Additionally placental melatonin acts with the MT1 and MT2 receptors and by the ROS (reactive oxygen species) to diminish placental oxidative damage [23,25]. Because melatonin maintains the turnover of the syncitiotrophoblast and protects the placenta from the antioxidant action, it has obtained the role of regulator of placental homeostasis [24].

In a recent case-control study by Ejaz H. et al., 2021, the serum level of melatonin and its major metabolite 6-OHMS in normal pregnant women was determined during each trimester of pregnancy and immediately after delivery (Table 2).

Blood samples were obtained from a cohort of 26 healthy pregnant women during each trimester of pregnancy, from 15 women scheduled for elective cesarean section (CS) before and after delivery, along with placental samples, and from 30 healthy non-pregnant women as controls. It showed that the levels of serum melatonin were significantly higher during pregnancy than in non-pregnant women and increased throughout pregnancy. In women undergoing CS, serum melatonin decreased markedly 24 h after delivery. Similar results were seen for serum levels of 6-OHMS, and placental tissue 6-OHMS levels correlated with the week of gestation at delivery.

In conclusion, maternal melatonin production is higher in pregnant than in non-pregnant women, increases significantly during pregnancy with the highest levels in the third trimester, and decreases abruptly after delivery [26]. These results suggest that the placenta is a major source of melatonin and supports a physiological role for melatonin in pregnancy. Because the placenta plays a key role in the numerous pathologies of pregnancy, its production of melatonin has also been studied in the context of high-risk pregnancy, as it is underlined in the review by Laste G. et al., 2021.

## 5. Melatonin in High-Risk Pregnancy

A high-risk pregnancy is defined as a pregnancy in which the mother or the baby may be at increased risk for health problems during the pregnancy and/or during and after delivery. A high-risk pregnancy may involve chronic health problems, such as diabetes or high blood pressure; infections; complications from a previous pregnancy, or other issues that might arise during pregnancy. [24]. The mechanisms involved in pregnancy complications have been studied, demonstrating the role of melatonin in these processes [23,24,26,27]. Particularly, the review by Last G. et al., 2021 reported that melatonin is important in gestational diabetes and preeclampsia.

### 5.1. Melatonin and Gestational Diabetes

Gestational diabetes mellitus (GDM) is one of the most common complications in pregnancy. Its previous definition was “any degree of glucose intolerance with onset or first recognition during pregnancy”. GDM is responsible for many risks for the mother, such as preeclampsia, primary cesarean delivery, preterm delivery, hydramnios, and for the child, including macrosomia, neonatal hypoglycemia, shoulder dystocia/birth injury, neonatal respiratory problems, hyperbilirubinemia, hypocalcemia, and intensive neonatal care. An additional long-term risk for the mother is a 30–70% recurrence risk of gestational diabetes, which is also largely dependent on weight gain between pregnancies, in addition to a 7-fold increase in the risk of diabetes occurrence after 5–10 years, as well as an increased risk in metabolic syndrome or cardiovascular disease [28].

In gestational diabetes, it was shown that a decrease in melatonin levels increased glucose transport to embryos, which could then augment the oxidative state of cells by decreasing free radical scavenging, increasing oxidative metabolism, or both. The formation of oxidative stress species would be a direct consequence of hyperglycemia, leading to various diabetic embryopathies [29]. In addition, recent reports demonstrated that melatonin receptor 1B (MTNR1B) gene polymorphisms may influence insulin secretion and pancreatic glucose sensing, causing gestational diabetes mellitus (GDM) [30,31]. The review by Laste G et al., 2021 analyzes 12 articles, of which the main findings are the association between MTNR1B polymorphisms and gestational diabetes mellitus. It cites one of the first reviews by Zhang and colleagues [32], which found a higher frequency of G alleles in MTNR1B rs10830963 and T alleles in both MTNR1B rs1387153 and rs1801278 in GDM patients than in healthy controls. Additionally, a meta-analysis by Huang and colleagues [33] indicated that the variant G allele of the MTNR1B rs10830963 polymorphism significantly increased the risk of GDM. Then, in Alharbi K.K. et al., 2019, MTNR1B variants were found to be related to insulin secretion and impaired β-cell function and GDM could develop when a genetic predisposition to pancreatic islet β-cell impairment is unmasked by increased insulin resistance during pregnancy. Finally, the review by Liao and colleagues [34] showed that MTNR1B is likely involved in the regulation of glucose homeostasis during pregnancy. MTNR1B rs10830963 and rs1387153 were shown to influence fasting plasma glucose (FPG) [32,35] and were associated with a higher risk of type 2 diabetes. In conclusion, this review states that decreased melatonin levels were found to be positively correlated with an increased risk of glycemic disorder, and melatonin administration was found to reduce the risk of glycemic disturbance [36].

### 5.2. Melatonin and Preeclampsia

Preeclampsia is an expressive gestational disorder that affects 3 to 10% of pregnancies, classically characterized by high blood pressure and proteinuria [37]. It is characterized by new-onset hypertension which usually occurs after 20 weeks of gestation and evidence of end-organ dysfunction. The end-organ disease resulting from preeclampsia is varied and can include proteinuria, acute kidney injury, hepatic dysfunction, hemolysis, thrombocytopenia, and, less frequently, liver rupture, seizures (eclampsia), stroke, and death. There are several risk factors for developing preeclampsia such as a history of preeclampsia in a prior pregnancy, diabetes, hypertension, obesity, and multiple pregnancies [38]. The placenta has always been a central figure in the etiology of preeclampsia because the removal of the placenta is necessary for symptoms to regress [39]. Impaired placenta function may cause preeclampsia, leading to oxidative stress and pro-inflammatory biomarkers. In preeclamptic placentas, the expression of melatonin-synthesizing enzymes, melatonin levels, and melatonin receptors is reduced [40]. It is plausible that melatonin, an antioxidant with anti-inflammatory and anti-apoptotic effects, protects cells/tissues by indirectly increasing gene expression and reducing errors in nuclear deoxyribonucleic acid (DNA) [41]. The review by Chuffa LGA et al., 2019 documents that melatonin levels and its receptor are depressed during severe preeclampsia. In the first trimester of gestation, the MT1 receptor is more important for promoting villous cytotrophoblast syncialization by protecting trophoblastic cells against oxidative injuries and increasing apoptosis in altered ones.

Experimental evidence supports the role of melatonin in providing adequate placental perfusion while preventing vascular damage, inflammation, and local oxidative stress. At pharmacologically relevant levels, melatonin reduces ROS (Reactive oxygen species) and hypertension in preeclamptic tissue and may be considered useful as a natural adjuvant for the treatment of preeclampsia. The review by Man GCW et al., 2017 states that melatonin inhibits the VEGF (vascular endothelial growth factor) expression and hypoxia-induced factor-1α(HIF-1α), a mediator of VEGF, which is predominantly active within the vascular endothelial cells. That the VEGF detrimentally influences the outcome of preeclampsia under the influence of the clock genes network is entirely speculative. As VEGF is predominantly active within the vascular endothelial cells, it lends itself as a prime candidate to this speculative, but plausible, association of a “clock” determining factor for preeclampsia. In fact, normal blood pressure is known to vary in a circadian manner, but in those with preeclampsia, this circadian relationship is lost, probably influenced by circadian rhythm under the melatonin. Hobson S.R. et al., 2018, observed that melatonin supplementation prolonged gestation and reduced the dosage of antihypertensive drugs, suggesting there is likely to be a new and promising paradigm shift in terms of diagnostics and therapeutics [42].

### 5.3. Melatonin and Intrauterine Growth Restriction (IUGR)

Fetal growth restriction (FGR) or IUGR is a common fetus and neonatal complication of preeclampsia [43]. In this pregnancy condition, the fetus does not reach its biological growth potential as a consequence of impaired placental function, which may be because of a variety of factors. Fetuses with FGR are at risk for perinatal morbidity and mortality and poor long-term health outcomes such as impaired neurological and cognitive development, and cardiovascular and endocrine diseases in adulthood. It is usually defined by the statistical deviation of fetal size after a population-based study; its algorithm was summarized in “Delphi procedure consensus criteria for defining fetal growth restriction”. SGA, instead, differs from FGR, principally because it also encompasses a majority of constitutionally small but healthy fetuses at lower risk of abnormal perinatal outcomes (Gordijn SJ., 2016).

Pregnancy complicated by hypertensive disorders, including preeclampsia, has been shown to substantially increase the risk of FGR in subsequent small-for-gestational-age newborns [43]. In some recent studies, it was established that in pregnant women with placenta insufficiency manifested as intrauterine fetal growth restriction syndrome, the blood concentrations of PlGF (Placental growth factor) significantly decrease. PIGF has a pro-angiogenic effect on the placental tissue; it also stimulates the proliferation of the trophoblast, and it is known as a predictor and a diagnostic marker of preeclampsia and is responsible for the angiogenesis in the placenta. Recently, it was found that in the case of IUGR, melatonin concentrations in maternal blood significantly decrease, resulting in strengthening of the pro-inflammatory immunity, increasing the levels of the anti-inflammatory cytokines, which was significant in the case of IUGR [44]. In the case-control study by Berbets A.M. et al., 2020, whether the level of melatonin, cytokines, and PlGF in umbilical blood after birth is different in the case of IUGR compared to normal fetuses was investigated. The study was conducted on 14 women whose pregnancies were complicated with IUGR. The control group consisted of 13 women who had uncomplicated pregnancies. The umbilical blood was taken immediately after a baby’s birth from the placental side of the clamped and cut umbilical cord before the birth of the placenta. In the end, it was confirmed that the level of melatonin and PIGF in the umbilical blood taken during the third period of labor from pregnant women whose pregnancies were complicated with IUGR is considerably decreased compared to the patients with uncomplicated pregnancies. This is probably caused by the placenta’s altered production of melatonin. Larger studies are needed to confirm this correlation and to confirm melatonin as a new marker of placenta function.

## 6. Melatonin and Labor

Uterine contraction is essential to progress into active labor for vaginal birth (Figure 2). In the last decade, induction of labor has been a common practice performed with the intent of reducing risks to the mother and/or baby by simply calling an end to the pregnancy. Ideally, induced labor progresses to vaginal delivery, but most induced labors fail and become a caesarean delivery [45,46,47]. Caesarean section increases the fetal and mother mobility, such as postpartum hemorrhage and venous thrombosis [48,49].

Maternal melatonin levels increase with advancing gestation, peaking during labor and then falling rapidly after birth [50]. The myometrium (uterine muscle) expresses the melatonin receptor MT2, and it is more highly expressed in laboring myometrium, collected at intrapartum caesarean section, than in myometrium from non-laboring women [51]. It has been proposed that melatonin receptor 1B (hMTNR1B) synergizes with oxytocin to promote nocturnal uterine contractions [7]. In fact, in humans, spontaneous labor in term pregnancies is more often initiated and more babies are born at night [52], a time when the pineal gland secretes the hormone melatonin into circulation. Melatonin receptor expression in the human pregnant uterus has been reported only during labor. In late-term pregnancy, circulating melatonin is of fundamental importance to induce timing and degree of contractions; conversely, its acute inhibition with light suppresses myometrial contractions [53]. In the same study, melatonin was also shown to increase the expression of the protein connexin, a gap-junction protein necessary for myometrial cell communication and the synchronization of uterine contractions. In addition, another study underlined that as melatonin increases, so does the sensitivity of the myometrium to oxytocin-induced contractions [7].

Taken together, these in vivo and in vitro observations suggest that melatonin plays a biological role in the timing of the onset of spontaneous labor and the effectiveness of spontaneous uterine contractions in labor [52]. The manuscript by Rahman SA et al., 2019, evaluated the impact of light-induced modulation of melatonin secretion on uterine contractions in women during the late third trimester (~36–39 weeks) of pregnancy in two inpatient protocols. The result of this study confirmed that there is a positive relationship between melatonin concentrations and uterine contractions in women after ~35 weeks of pregnancy. Moreover, there are many potential applications of this discovery to provide a new mechanism for therapeutically influencing the timing of labor and childbirth, since endogenous melatonin levels can be suppressed by both light and pharmacologic agents, and melatonin receptors can be activated by melatonin. Regarding this interesting topic, there is a work in progress, a double-blind randomized placebo-controlled trial MILO (Melatonin as an adjuvant agent in the induction of labor), by Swarnamani K et al., 2021, that aims to test the hypothesis that the inclusion of melatonin will reduce the need for the caesarian section in induced labor delivery [9].

## 7. Melatonin and Preterm Birth

Preterm delivery survivors often present long-term neurodevelopmental sequelae, such as motor, learning, social-behavioral and communication difficulties. Melatonin can be considered a primary candidate for conferring neuroprotection in preterm infants. This hormone and its metabolites are extremely useful antioxidants and free radical scavengers. Melatonin has already been shown to act as a neuroprotectant in adult cerebral ischemia [54] and an increasing amount of studies have consistently demonstrated that melatonin protects the developing brain by preventing abnormal myelination and inflammatory glial response [55], which are the key causes of white matter damage [56]. Preterm infants are deprived of the maternal melatonin they would usually have received in the last weeks/months of pregnancy and their endogenous making of the hormone may be delayed even by months [57]. Biran V. et al., 2019, theorized that melatonin circulating levels would be lower in preterm compared to term infants. They showed a prospective, longitudinal, multicenter study to assess melatonin, and 6-sulfatoxy-melatonin (aMT6s) concentrations. Among 209 neonates engaged, 110 were born earlier than 34 gestational weeks (GW) and 99 were born after 34 GW. Plasma melatonin concentrations measured at birth and on Day 3 had lower detectable levels (≤7 pg/mL) in 78% and 81%, correspondingly, of infants born before 34 gestational weeks compared to 57% and 34%, respectively, of infants born after 34 gestational weeks. The circulation of plasma melatonin concentrations was found to be linked with gestational age at both time points. Average urine aMT6s concentrations were considerably inferior in infants born before 34 gestational weeks, both on Day 1 (230 ng/L vs. 533 ng/L) and Day 3 (197 ng/L vs. 359 ng/L). In conclusion, melatonin secretion appears very low in preterm infants, providing the rationale for testing supplemental melatonin as a neuroprotectant in clinical trials [58] Consequently, exogenous melatonin management during the first weeks of extrauterine life could provide support in preventing/mitigating the negative results of prematurity on brain development. Additionally, the pharmacokinetics of melatonin in preterm infants [58] show prolonged half-life and clearance and a reduced size of distribution. The Carloni S. et al. 2017 study administered oral melatonin to 15 preterm infants in the following amount regimens: one intragastric bolus of 0.5 mg/kg and three intragastric boluses of 1 mg/ kg at 24-h intervals, and three intragastric boluses of 5 mg /kg at 24-h intervals. Regarding the pharmacokinetic outcomes, the half-life in plasma ranged from 7.98 to 10.94 h, the area under the curve ranged from 10.48 to 118.17 mg/mL/h, and the period to obtain the maximum concentration ranged from 2.91 to 4.70 h. This study also determined that the pharmacokinetic profile of melatonin in premature infants is more diverse than in adults, with a protracted half-life and time to maximum concentration, and that a single oral melatonin dose repeated every 12 or 24 h could be used to take and maintain high serum concentrations for therapeutic purposes in preterm infants. Few studies in very preterm infants have tried to assess the capacity of melatonin to defend the brain against oxidative stress-induced damage caused by preterm birth. The prospective, multicenter, double-blind, randomized vs placebo study carried out by Garofoli F. et al., 2021, aims to observe the neurobehavioral expansion of infants throughout the first 2 years of life. They focus on clinical signs and symptoms and objective instrumental findings, comparing melatonin-treated patients with placebo-treated patients. To determine whether melatonin supplementation can be considered a neuroprotective action against brain damage, they try to define which patients and disabilities benefited the most from melatonin supplementation and which the least. This study could be a phase towards making a preventive treatment tailored to a single patient’s characteristic birth weight, such as gestational age, sex, or other variables and illnesses related to preterm birth. In the end, this analysis aims to assess the potential neuroprotective role of melatonin in very preterm newborns. Melatonin administration could perhaps be an effective means of mitigating neurological impairments, strengthening the complex supportive therapy already provided in Neonatal Intensive Care Units. Furthermore, its protective effects might also help to decrease national health system costs related to the rehabilitation of developmental disabilities due to preterm birth.

## 8. Melatonin in Breastfeeding

Breastfeeding is strongly suggested, with benefits for both infant and mother. The American Academy of Pediatrics recommends breast milk as the best source of enteral diet for preterm infants. Breastfeeding rates are variable across nations as well as across different ethnic groups within a country. Breastfeeding affords defense against a wide variety of medical illnesses that may begin at different time points over an individual’s lifespan, including hospital admissions for respiratory infections and neonatal fever [59], offspring childhood obesity and cancer [60] and sudden infant death syndrome [61]. The benefits of breastfeeding are studied, specifically, in the setting of the impact of breast milk on the infant’s gut, and thereby on the progress of the gut-brain axis and the immune system. It has been verified in vitro and in animal model systems that all these bioactive components of human milk not only control the development of the intestinal mucosal barrier but also have the capacity to stimulate the healing and repair procedures in the injured intestinal epithelium [62]. The melatonergic pathway may be closely associated with gut regulation as well as the procedures driving the immune-inflammatory relations of the gut–brain axis. Such interactions of the melatonergic pathways and immune-inflammatory processes form the backdrop of the gut–brain axis upon which the elements of breast milk will act. The part of the melatonergic pathways in these processes is highlighted. It should be noted that infants do not show a circadian production of melatonin until they are 3–5 months old, which is about the time corresponding to the termination of breastfeeding for many women, perhaps suggestive of an adaptive reaction in infants that are no longer obtaining melatonin from breast milk, or even an indicant that infants may not necessitate melatonin until this period. In numerous non-western nations, breastfeeding is continued until the infant is aged 12 months and beyond, representative that the demands on women in Western culture are likely to support their earlier termination of breastfeeding, rather than being an evolutionary-derived process. The night-time intensification in pineal melatonin increases circulating maternal melatonin concentrations, which are shifted in the breast milk to the sucking infant, along with N-acetyltrans-ferase to N-acetylserotonin (NAS) and several melatonin metabolites [63]. Night-time breast milk has superior levels of melatonergic pathway products that may, among other properties, act to entrain the infant’s evolving circadian rhythms. Night-time breast milk is consequently likely to have higher antioxidant, anti-inflammatory and immune controlling effects [64], involving arising from the impact of breast milk melatonin on the infant’s developing microbiome and gut penetrability, with effects for immune system progress, which is suggestively controlled by circadian factors [65].Given the status of the gut microbiome to an array of childhood and adult onset illnesses [66], including metabolic dysregulation, night-time breast milk melatonin is probable to be of some meaning to the etiology of a varied array of medical conditions. Maternal stress regulates many breastmilk factors [67], and, as well as gut penetrability, melatonin will be significant to the inhibition of these stress results, both in the mother and infant. The function of melatonin in night-time breast milk, and its effect on the supplementary mechanisms of breast milk, immediately requires more research. Recently, Anderson G. et al. suggested that melatonin should be added to a night-time precise formula feed in order to bring formula feed nearer to the benefits of breast milk [61]. Another study [68] relating melatonin levels in the breast milk of five women and three artificial formulations discovered that melatonin followed a clear circadian pulse in breast milk but was undetectable in all artificial formulas. They also applied a questionnaire to 94 mothers, finding that totally breastfed infants had a significantly lower rate of colic attacks and a lower gravity and incidence of irritability attacks than formula-fed infants. The breastfed infants also trended toward a longer nocturnal sleep period. Joining the questionnaire and the milk/formula analysis results, the authors proposed that the reason that breastfeeding was more beneficial than formula regarding infantile colic may be the role breast milk plays in melatonin absorption through the infant’s gastrointestinal tract. The supplement of melatonin to formula feed could be most helpful in premature infants, paralleling the common supplement of fats, proteins, and carbohydrates to breast milk for babies born preterm [69].

## 9. Melatonin in Newborn

Melatonin is a highly effective antioxidant and free radical scavenger, and it has anti-inflammatory properties. In children and neonates, melatonin has been used generally, including for respiratory distress syndrome, bronchopulmonary dysplasia, periventricular leukomalacia (PVL), hypoxiae ischemia encephalopathy and sepsis. In addition, melatonin can be used in childhood sleep and seizure illnesses, and in neonates and children receiving surgery [70]. Newborns are mostly inclined to oxidative stress, which can be accounted for by the fact that neonates are simply exposed to high oxygen concentrations, have infections or inflammation, have reduced antioxidant defense mechanisms, and have high levels of free iron that are requisite for the Fenton reaction and lead to the production of the highly toxic hydroxyl radical. In this context, Saugstad coined the phrase “oxygen radical diseases” of neonatology [71] Hypoxiaeischemia (HI), an insult resulting from a lack of oxygen (hypoxia) or reduced perfusion (ischemia) in various organs, is a common cause of brain damage in neonates. Neonatal HI is the major cause of neonatal brain injury, resulting in cerebral palsy, learning disabilities, visual field deficits, and epilepsy. Carloni S. et al. examined whether melatonin administration before or after HI in immature rats has an excellent and long-lasting benefit on ischemic outcomes and found that melatonin could represent a potentially safe approach to perinatal brain damage in humans [72]. Moreover, they found that melatonin has long-lasting beneficial effects on HI occurring during the substantial neurologic developmental process. Signorini et al. demonstrated that HI induces an increase in desferrioxamine (DFO)-chelatable free iron in the cerebral cortex, which can induce cerebral oxidative stress, whereas the cerebral damage by the oxidative stress may be prevented by melatonin treatment. From the above studies, melatonin may be considered a treatment modality for reducing oxidative stress injury at HI and further preventing the ongoing neurological injury process, which may lead to developmental disabilities. Neonatal stress has been demonstrated as one of the most important factors affecting lifelong health [73] Neonatal stress in humans can originate from maternal postpartum depression, family conflict, or physical abuse. In a rodent study, Rasmussen D.D. et al. showed that daily melatonin supplementation suppressed hypothalamic gene expression of proopiomelanocortin which encodes the adreno-corticotropic hormone [74]. Munoz-Hoyos A. et al. studied 112 newborns, including normal term babies, preterm infants born before the 38th week, and babies with fetal distress. They found that neonatal stress can increase nocturnal melatonin production compared with normal term and preterm neonates [75]. Saito et al. showed that melatonin could modulate the activity of the hypothalamo-pituitary-adrenal (HPA) axis in chicks. Melatonin blocked the enhanced corticosterone concentrations observed after layers were either exposed to a novel, open field or injected intracerebroventricularly with corticotropin-releasing factor. This study provides direct evidence that melatonin plays a role in the regulation of the stress response in neonatal chicks. [76].

## 10. Hypoxic Ischemic Encephalopathy

Hypoxic ischemic encephalopathy (HIE) is a one of the causes for morbidity and mortality in newborns, although antenatal and neonatal care have been improved in recent years [77]. There are several causes behind HIE, acutely or chronically, during the prenatal (hypotension, severe hypoxia or infection), perinatal (cord occlusion or prolapse, abruption or placental insufficiency, uterine rupture) or postnatal stages (shock, respiratory or cardiac arrest) [78]. The effects of HIE can be different in preterm or term infants. In preterm infants, HIE mainly results in white and gray matter damage in diverse brain areas. As pre-oligodendrocytes are the dominant cell types in the brains of preterm infants, HIE during the perinatal period primarily affects these cells, which causes white matter injury in the brain [79,80]. The main brain damage patterns in term infants due to HI include basal ganglia and thalamus. Because the bilateral perirolandic cortex and central gray nuclei are injured, more severe encephalopathy and seizures occur [81]. The time of damage and timing of care play important parts in the pathogenesis of hypoxia-ischemia. The oxidative metabolism, inflammation, and continuation of the activated apoptotic cascades take place in the latent stage of the damage (in between 1 to 6 h) [82]. Six to 48 h after HI damage, a reduction in phosphate supplies and the release of excitatory neurotransmitters and free radicals occur in the second energy failure stage [83]. In the tertiary stage, months after acute ischemia, late cell death, remodeling of the damaged brain and astrogliosis occur [84]. Melatonin has neuroprotective properties against hypoxic-ischemic brain injury in animal models: Antiapoptotic action such as inhibition of cytochrome c release and reduced or blocked caspase-1 and caspase-3 activation improved the expression of anti-apoptotic proteins Bcl-2 and Bcl-xL, diminished Bad and Bax pro-apoptotic proteins, inhibited poly-ADP-ribose-polymerase cleavage, stimulated mitochondrial biogenesis and promoted the electron transport chain, upregulated antioxidant pathways and reduced lipid peroxidation [70,85,86]. Melatonin activity in the central nervous system increases the number of neurons in melatonin-treated animals in the CA1, CA2–CA3 areas, and dentate gyrus of the hippocampus and parietal cortex. Melatonin decreases the expression of the glial fibrillary acidic protein and reduces the expression of myelin basic protein and the role of oligodendrocytes (regulation of myelination process) [86,87]. Recent animal studies confirm that melatonin decreases intracerebral cellular inflammatory reaction and protects neurons against ischemic damage by reducing oxidative stress, lipid peroxidation, and radical oxygen species generation [59,88]. Carloni et al. have discovered positive results of melatonin administration before and after hypoxia in immature rats. They exposed in a neonatal rat brain increased expression and activity of SIRT1 (silent information regulator 1), reduced expression and acetylation of p53, and increased autophagy activation [89]. In the experimental study performed by Yawno et al. in a preterm fetal sheep, an increased oligodendrocyte cell number within the periventricular white matter and improved CNPase+ myelin within the subcortical white matter were found [90]. According to Alonso-Alconada et al., the treatment with melatonin at 15 mg/kg after neonatal hypoxia-ischemia led to reduced cell death, white matter demyelination, and reactive astrogliosis in rat pumps [91]. Signorini et al., 2009 [61] examined the use of melatonin in a rat pump model of hypoxic-ischemic encephalopathy and proved that hypoxia may produce the formation of desferrioxamine-chelatable free irons in the cerebral cortex and, in consequence, rise oxidative stress. The latter can be prevented by melatonin administrated before starting the ischemic procedure. Robertson et al. in a piglet study proved that a combination of melatonin (5 mg/kg) and hypothermia (33.5C) is more successful than any of these forms of the treatment alone [92]. Another possibility is to administer melatonin antenatally in order to prevent or reduce brain hypoxic insult in preterm babies. In the literature, there are not enough in vivo data to evaluate the real efficacy of melatonin in the damage caused by hypoxichemic encephalopathy. Potential randomized trials in preterm infants should combine melatonin with other aspirant neuroprotection molecules (steroids sulphate or magnesium).

## 11. Conclusions

In the last decade, a number of studies have focused on melatonin, discovering the different properties that establish its relevance to human health. From the studies selected in this review, new knowledge on the roles of melatonin emerged: modulator of circadian rhythm, which is important for neurodevelopment in the fetus; anti-inflammatory function against high-risk pregnancy; and stimulation of labor, which is fundamental for the success of vaginal delivery. In addition, the studies confirm its ability to cross the placenta and its role as a component in breastmilk for infants’ developing circadian rhythms, as well as after birth, and its ability to reduce oxidative stress in newborns. Although several studies have confirmed that melatonin supplementation in pregnancy and immediately after has a positive effect on the mother and fetus, long-term clinical trials are imperative to reach clinical outcomes that serve as a final consensus about the use of melatonin as a treatment in pregnancy and newborns. This suggests that the use of melatonin in early life could preserve the safety of life in the future.

## Figures and Tables

**Figure 1 biomedicines-10-03252-f001:**
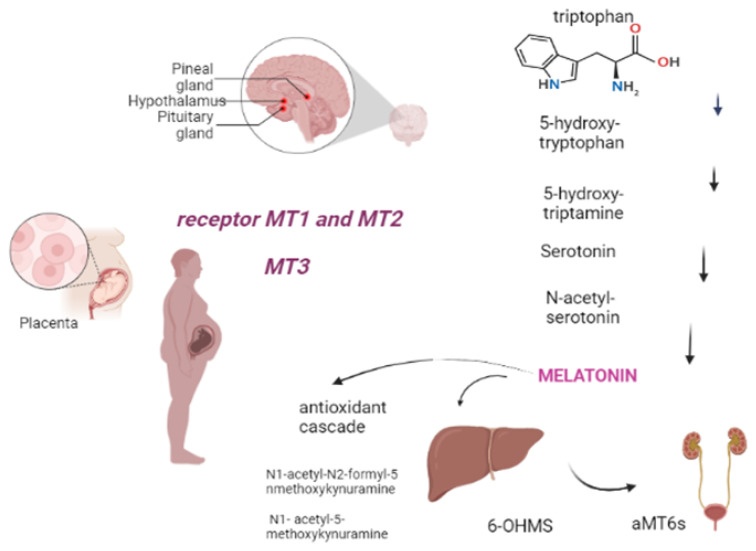
Melatonin comes from tryptophan, produced in the pineal gland after five stages of chemical reaction. It is subsequently metabolized in the liver and kidney or non-enzymatically (antioxidant cascade). Tissue targets include, during pregnancy, the placenta through the MT1, MT2 and MT3 receptors (Created with BioRender.com on 1 January 2022).

**Figure 2 biomedicines-10-03252-f002:**
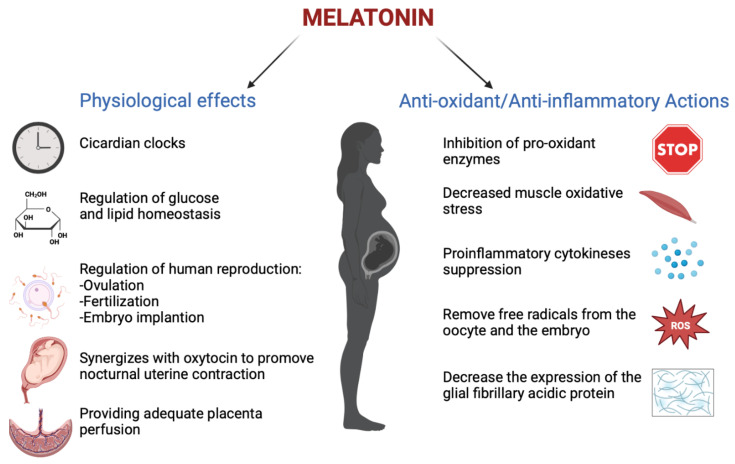
The physiological functions performed by melatonin in pregnancy (Created with BioRender.com on 1 January 2022).

**Table 1 biomedicines-10-03252-t001:** The Studies of melatonin cited.

Authors and Year	Type of Study	Aim of the Study	Outcomes and Conclusion
Tya Vine et al., 2022	Review	To determinate the influence of melatonin on sleep during pregnancy	Contrary to what animal studies have suggested, evidence from clinical studies to date suggests that melatonin use during pregnancy and breastfeeding is probably safe in humans. This review further emphasizes the need for clinical studies on sleep disorders, including exogenous melatonin, during pregnancy and lactation.
Ejaz H. et al., 2021	Case controlled trial	To determine serum levels of melatonin and its major metabolite 6- hydroxymelatonin sulfate (6-OHMS) in normal pregnant women during each trimester of pregnancy, and immediately after delivery	The role of the placenta in melatonin production may lead to the use of melatonin as a preventative treatment for obstetric conditions.
Garofoli F et al., 2021	Randomized controlled trial	The planned study will be the first aiming to evaluate the capacity of melatonin to mitigate brain impairment due to premature birth.	The trial protocol refers to version v4–11/2019. Recruitment was expected to start in April 2020, but the current Coronavirus Disease-19 emergency delayed it.
Laste G. et al., 2021	Review	Investigating the relationship between melatonin and high-risk pregnancy.	The association between melatonin receptor 1B polymorphisms and gestational diabetes mellitus is the most common physiological mechanism relating The circadian rhythm, decreased melatonin production, and anti-inflammatory and antioxidant effects were explored
Berbets AM et al., 2020	Case control	To investigate whether the level of melatonin, cytokines, and PlGF in umbilical blood after birth is different in the case of IUGR compared to normal fetuses.	The concentrations of melatonin and PlGF in the umbilical blood during labor are significantly lower in the case of IUGR compared to normal pregnancies
Swarnamani K et al., 2020	A double-blind randomized placebo-controlled trial	to test the hypothesis that addition of melatonin will reduce the need for caesarean section.	Following completion of the trial, it is intended that the aggregated results will be published in peer-reviewed. The clinical trial will be carried out.
Biran V. et al., 2019	prospective cohort study	To determine serum levels of melatonin in different GW	Melatonin production and excretion are significantly lower in preterm infants born before 34 GW when compared to more mature infants, based on intrinsic properties of this molecule.
De Almeida Chuffa et al., 2019	Review	To focus on the main in vivo and in vitro functions of melatonin on uterine physiological processes, such as decidualization and implantation, and also on the feto-maternal tissues, and reviews how exogenous melatonin functions from a mechanistic standpoint to preserve the organ health.	The benefits of melatonin during the pregnancy are indisputable (especially during the last trimester of gestation where it improves P4 synthesis and inhibits premature oxytocin release). It has been consistently shown that melatonin works to control the pathogenesis of neonatal morbidities and diseases associated with inflammation, cell death, and oxidative stress
Genario R. et al., 2019	Review	To further investigate the evidence available on the effects of melatonin supplementation in animal and human studies, focusing on its potential application to gynecology.	Melatonin supplementation may offer a therapeutic as well as preventative potential in the area of gynecology and obstetrics due to its antioxidant properties and activity as a hormone modulator.
Ji Yeon Lee MD et al., 2019	Case controlled trial	To investigate the effects of maternally administered melatonin on preterm birth and perinatal brain injury in a mouse model of maternal inflammation.	This study revealed that melatonin decreased the incidence of preterm birth and perinatal brain injury in a mouse model of LPS-induced maternal inflammation.
Rahman SA et al., 2019	Randomized case control	To evaluate the impact of light-induced modulation of melatonin secretion on uterine contractions in women during late third-trimester	A new mechanism was provided for therapeutically influencing the timing of labor and childbirth, since endogenous melatonin levels can be suppressed by both light and pharmacologic agents, and melatonin receptors can be activated by melatonin.
Carlomagno G. et al., 2018	Review	Positive effects on the outcomes of compromised pregnancies	Melatonin has a cytoprotective effect, together with the immunomodulatory one essential for the success of pregnancy and correct fetal development. To better define the posology of melatonin in order to achieve the most effective responses, both for the mother and the fetus.
Hobson SR et al., 2018	Original Manuscript	To determine if melatonin could be a useful adjuvant therapy	It was shown that melatonin is safe to take in pregnancy and while it does not look promising as a preventative therapy, our observations suggest that it is worth exploring further as an adjuvant therapy in women with established preeclampsia, particularly those with early-onset preeclampsia.
Gene Chi Wai et al., 2017	Review	To provide an overview, from an immunological perspective, on the roles of circadian clock and melatonin in pregnancy.	Introduction of exogenous melatonin might have multiple beneficial effects on protecting the mother and fetus toward immunocompromising the effect of exogenous melatonin on immunity during pregnancy remains to be elucidated in pregnancy.
Katzer D. et al., 2016	Case controlled trial	To evaluate the antioxidative capacity in breast milk and its regulation by time of day	In human preterm breast milk, not only melatonin but also Gpx3, SOD, and TAOC showed a diurnal rhythm, with a night-time maximum level for melatonin and Gpx3 and a daytime maximum level for SOD and TAOC
YU WEN LIN. et al., 2018	Case controlled trial	They evaluated whether melatonin inhibits ER stress in cultured neurons exposed to oxygen and glucose deprivation and in rats subjected to transient focal cerebral ischemia	Their results demonstrated that animals treated with melatonin had significantly reduced infarction volumes and individual cortical lesion sizes as well as increased numbers of surviving neurons Melatonin reduced brain infarction following Middle Cerebral Artery occlusion at 24 h

**Table 2 biomedicines-10-03252-t002:** This table shows melatonin levels characteristic for pregnancy, newborns and mother`s milk. The melatonin plasma level increases significantly during pregnancy with highest levels in the third trimester and decreases abruptly after delivery. Subsequently, it is secreted in the breastmilk and its concentration in the serum of newborn becomes low after early days.

Serum melatonin concentration	
I TRIMESTER OF PREGNANCY	Up to 611.4 pg/mL
II TRIMESTER OF PREGNANCY	Up to 1246.4pg/mL
III TRIMESTER OF PREGNANCY	Up to 1372 pg/mL (Ejaz H. et al., 2021)
AT DELIVERY	Up to 158 pg/mL (Biran V., 2019)
Serum melatonin concentration	
NEWBORN AT BIRTH	Up to 184 pg/mL
NEWBORN ON 3 DAYS AFTER BIRTH	Up to 75 pg/mL (Biran V., 2019)
Milk melatonin concentration	
MOTHER MILK	Up to 20 pg/mL (Biran V., 2019)

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
