# Peer review of "The Role of Melatonin in Pregnancy and the Health Benefits for the Newborn"

_biomedicines, 2022, doi:10.3390/biomedicines10123252_

Round 1

Reviewer 1 Report

This is an interesting review concerning potential role of melatonin in pregnancy.

1. The paper need however extensive correction. The manuscript contains multiple “short cuts” and sentence with missing parts (probably fragments removed during editing). E.g.:

Page  5.“Melatonin is largely metabolized in the liver and kidneys by P450 monooxygenases, followed by conjugation of the resulting 6-sulfa-toxy-melatonin to produce the main metabolite, 6-sulfatoxymelatonin, in the urine (Hardeland R., 2009)”

This also lead to unjustified conclusions (production in urine?). 

Page 6 Lines 19-21. MT3 (quinone reductase II) was considered as intracellular receptor for melatonin and most probably does not for dimers with MT1 or MT2.

2. Manuscript also requires extensive correction by native speaker.

3.  Please use consequently the same nomenclature for chemical compounds.

6-sulfa-toxy-melatonin, 6-sulfatoxymelatonin, 6-hydroxymelatonin sulfate

and 6-sulfatoxy-melatonin (aMT6s)!

4. Authors Affiliations are missing.

5. Please provide proper description of tables:

Page 2 “Table 1. This is a table. Tables should be placed in the main text near to the first time they are cited”

6. I do not think Table 1 should be presented in the introduction. It contains very interesting data which also require careful discussion.

7. The organization of the manuscript has to be rethought. There is a lot of repetitions.

8. Whole chapter 4 (melatonin in placenta) is based on one paper. Please provide some more information, which cells produce melatonin, which cells are its targets etc.

9. Please check and correct citation style throughout the manuscript.

10. Please switch paragraphs 5 and 6 (first labor than preterm birth).

11. The sentence “Such interactions of tryptophancatabolites (TRYCATETs) with the mel-atonergic pathways” makes no sense because tryptophan is the mayor substrate for melatonin, thus is a TRYCATETs.

12.  It would be good to add the table with melatonin levels characteristic for pregnancy, newborns, infants and mother`s milk

14. Please explain NAS

15 Please start the description of Hypoxiaeischemia with new paragraph (and use term hypoxia-ischaemia).

16. Overall I have a feeling that whole review is based on other reviews but in fact it should be focused on recent research papers.

Author Response

Response to Reviewer 3:

I describe point to point the modification of the manuscript.

  1. We corrected the major multiple “short cuts” and sentence with missing. We underlined each modification by the comments.
  2. I couldn’t correct by native speaker cause of enough time to work.
  3. We correct the nomenclature of chemical compounds.
  4. Authors Affiliations is Department of Obstetrics and Gynecology, Azienda Ospedaliero-Universitaria S. Anna, University of Ferrara, Cona, 44122 Ferrara, Italy.
  5. We corrected the the table 1 and new table 2
  6. We discussed the tables.
  7. we replaced the major repetitions.
  8. we provide some more information about chapter 4.
  9. We checked and corrected citation style throughout the manuscript.
  10. We switched paragraphs 5 and 6 (first labor than preterm birth).
  11. The corrected the sentence about TRYCATETs .
  12. We added the table with melatonin levels characteristic for pregnancy, newborns, infants and mother`s milk.
  13. We explained NAS

15 We started the description of Hypoxiaeischemia with new paragraph (and use term hypoxia-ischaemia).

16.In this work we tried to select the last literature works about melatonin, and we also used the reviews because there aren’t enough experimental studies concluded about the topic of the work. However, we focused on some research articles included in the reviews.

Lastly we corrected the email address [email protected].  

Reviewer 2 Report

This manuscript is a review for melatonin, which controls crucial physiological processes of circadian rhythms, immune, and others. The role of melatonin in pregnancy is written well with many current papers. Antioxidant function and regulation of circadian rhythms, main roles of melatonin, are described well in pregnancy. 

If possible, physiological functions performed by melatonin in pregnancy should be shown illustrated to be easy to understand as shown in the previous paper  (The Journal of Physiological Sciences, 2021, 71:27, doi 10.1186/s12576-021-00812-2).

Author Response

We have made a picture about physiological functions performed by melatonin in pregnancy. 

Sorry for late answer. 

Drs Milano and Pierdomenico

Reviewer 3 Report

1The Review entitled: "The role of melatonin in pregnancy and the health benefits of the newborn" analized the melatonine role at the begining of human life, in pregnancy and in fetus. The paper is well structured and very interesting, considering the different beneficial properties of melatonin on human health and quality of life. I think that this Rewiew is suitable for its publication in Biomedicines (ISSN 2227-9059) after minor revisions:

1.      Much of the bibliography is not recent. Justify the use of old references.

2.      In the section “2. Melatonin’s metabolism” the authors could add a figure to schematize which gland produces melatonin, its formula, its precursor, and its metabolites, so as to make more immediate, what is specified in section 2

3.      or they could include a generic summary figure with the chemical formula, its precursor, the gland that produces it, and the positive or negative effects in pregnancy.....etc

Author Response

In the section 2 we have inserted the picture about melatonin's metabolism.

Sorry for late answer. IT is underlined by color green.

Drs Milano and Pierdomenico